# Methodology of Studying Effects of Mobile Phone Radiation on Organisms: Technical Aspects

**DOI:** 10.3390/ijerph182312642

**Published:** 2021-11-30

**Authors:** Katerina Bartosova, Marek Neruda, Lukas Vojtech

**Affiliations:** Department of Telecommunication Engineering, Faculty of Electrical Engineering, Czech Technical University in Prague, Technicka 2, Prague 6, 166 27 Prague, Czech Republic; katerina.bartosova@fel.cvut.cz (K.B.); lukas.vojtech@fel.cvut.cz (L.V.)

**Keywords:** dosimetry, DNA, electromagnetic radiation, health effects, methodology, mobile phones, technical aspects

## Abstract

The negative influence of non-ionizing electromagnetic radiation on organisms, including humans, has been discussed widely in recent years. This paper deals with the methodology of examining possible harmful effects of mobile phone radiation, focusing on in vivo and in vitro laboratory methods of investigation and evaluation and their main problems and difficulties. Basic experimental parameters are summarized and discussed, and recent large studies are also mentioned. For the laboratory experiments, accurate setting and description of dosimetry are essential; therefore, we give recommendations for the technical parameters of the experiments, especially for a well-defined source of radiation by Software Defined Radio.

## 1. Introduction

With the rapid development of wireless communication, the exposure of people to non-ionizing electromagnetic radiation, especially from mobile phones, is growing rapidly, raising concerns about its possible effects on human health.

This paper deals with the main problems of experiments studying the effects of mobile phones on human health, namely on cells and DNA. We focus on scientific methods, refer to their advantages and difficulties and give recommendations for future research, especially in the technical aspects of the experiments.

### 1.1. Types of Studies of Biological Effects

From the possible effects of radiofrequency radiation on the human body, the most studied are the effects on the brain, cancer incidence, and fertility, including induction of certain enzymes, neurological symptoms, toxicological effects, genotoxicity, carcinogenicity, and decreasing the fertilizing potential of sperm cells. The published literature studies the influence of radiofrequency radiation on mitochondria, apoptosis pathways, heat shock proteins, free radical metabolism, cell proliferation, cell differentiation, DNA damage, the plasma membrane, etc. [1,2,3,4,5].

The biological effects of radiofrequency radiation can be researched by three basic types of studies: laboratory studies in vivo, in vitro, and epidemiological studies, which will not be further discussed in this paper.

#### 1.1.1. Laboratory Studies In Vivo

In in vivo research methods, laboratory animals are exposed to radiofrequency radiation, and the effects of the exposure are evaluated, e.g., samples of cells or tissues are collected to study possible damage. The laboratory studies allow better control of the experimental conditions compared to epidemiological studies and help to answer basic questions. The problem of these studies is often the number of samples, which in many cases is not high enough to make a statistically significant conclusion. Furthermore, the application of the results on humans is always problematic [6]. Zhu et al. [7] suggest that larger animals (e.g., rabbits) are more similar to humans in many aspects (e.g., skull thickness) and, therefore, more suitable for radiofrequency radiation experiments. Comparisons of in vivo experiments are mostly difficult because many different experimental settings are used [8].

#### 1.1.2. Recent Large In Vivo Studies

Recently extensive rodent toxicology and carcinogenesis studies were carried out in the US National Toxicology Program (NTP). Two different types, or modulations, of radiofrequency radiation (GSM and CDMA) commonly used in mobile phone networks in the United States, were used to expose rats and mice to identify potential toxicity or cancer-related hazards. Animals were exposed to radiofrequency radiation for 18 h per day with 10 min off/on cycles, 7 days per week, for up to 2 years. There were 90 male and 90 female animals in each group. Whole-body SAR exposures of 0, 1.5, 3, or 6 W/kg at 900 MHz were used for rats. Whole-body SAR exposures of 2.5, 5, or 10 W/kg at 1900 MHz were used for mice [9,10]. The doses of radiation in the studies were generally higher than those used in mobile phones. The lowest exposure level used in the studies was equal to the current local tissue exposure limit.

A genetic toxicology study with analysis of the comet assay and micronucleus assay was also performed (for five different tissues per animal) [9,10].

Follow-up studies by the NTP to investigate mechanisms of genetic damage are underway [11]. For future studies, NTP wants to focus on developing measurable physical indicators, or biomarkers, of potential effects of radiofrequency radiation. These may include changes in metrics, such as DNA damage in exposed tissues, which can be detected much sooner than cancer [12].

Another large recent study was performed by the Ramazzini Institute in Italy. Rats were exposed for 19 h/day to a 1.8 GHz GSM far-field of 0, 5, 25, or 50 V/m (SAR up to 0.1 W/kg) from the prenatal period until natural death [13].

Both NTP (simulating near field exposure from a mobile phone) and the Ramazzini study (simulating far-field exposure from a base station) used simulated radiofrequency signals emitted by generators. The strength of the studies is that they exactly controlled how much radiofrequency radiation the animals received [12]. The disadvantage of these studies (from the technical point of view) is the use of a signal generator, i.e., the absence of the unpredictable changes of the radiofrequency signal (in contrast to the real phone signal, as discussed in Section 2.). Other general possible shortcomings of the NTP study are discussed in [14] and shortcomings of Ramazzini and other studies in [15].

#### 1.1.3. Laboratory Studies In Vitro

In vitro studies usually use different types of cells that are exposed to radiation in a special container. After, the exposure of the damage to the cells is evaluated (e.g., their viability and DNA damage). Therefore, for in vitro experiments, comparison or reproducibility is a challenge. This is not only due to different experimental settings but can be caused even by slight modifications of the evaluating methods [8,16].

## 2. Experiments Investigating the Effects of Radiofrequency Radiation on Cells and DNA

Carcinogenic growth is mostly initiated by damage to the cell’s genome, and therefore, many studies have investigated the effects of electromagnetic fields on DNA and chromosomal structure [16].

### 2.1. Experimental Settings

The published experiments studying the effects of radiofrequency radiation differ in the type of investigated cells, frequency and intensity of the electromagnetic radiation, time of exposure, and methods evaluating the effects of exposure on cells and DNA. The most important parameters of the experiments are summarized below [8,16,17,18,19,20,21,22]:Method

Both in vitro and in vivo methods are used.

Cells

The studies use different types of cells. The cells may originate either from animals, mostly laboratory mice and rats, or humans. The cells can be freshly collected (e.g., lymphocytes, leukocytes, sperm, or skin cells) or cultured (e.g., brain tumor, lung, skin, or stem cells).

Specific Energy Absorption Rate (SAR)

The value of SAR in published works reaches mostly from <1 to 5 W/kg, higher SAR values are less frequent. In some papers, the SAR value is not given at all.

Frequency

Different frequencies are used, most works use frequencies ≤2000 MHz, and some are around 2500 MHz. Higher frequencies are rarely used.

Other parameters of the radiation

Power intensity of the radiation and exposure time are crucial parameters. Other important characteristics of electromagnetic radiation are variability, modulation (continuous or pulse-modulated waves), or shape of the waves (square or sine waves).

Sources of radiation

Many studies use usual mobile phones to expose the samples or animals, often alternation of standby mode and dial mode is used. It is also obvious that the generation of mobile technology used, e.g., 2G (GSM), 2,5G (GPRS), 3G (LTE), plays a role. Another possibility is a source of radiofrequency radiation, where exact parameters (e.g., frequency) can be set.

The above-mentioned parameters are very important, and even slight variations can lead to discrepant results in seemingly similar studies [16]. According to Saliev et al. [19], the most important parameters are the used cell line and the type of radiation (frequency, but also modulation and shape of the waves). Moreover, the conditions in cell culture incubators should be controlled carefully. A study by Mild et al. [23] detected enhanced levels of extremely low-frequency magnetic fields in cell culture incubators, which can have a significant influence on the cell cultures.

A review by Panagopoulos [24] comes to an important conclusion that once the electromagnetic field is polarized, includes extremely low frequencies, and has adequate intensity, then the main parameter is variability. It suggests the extreme and unpredictable variability of the real mobile telephone signals seems to be the main reason for the corresponding bioactivity.

### 2.2. Dosimetry

Exposure to a uniform electromagnetic field results in a non-uniform deposition and distribution within the body and, therefore, a dosimetric approach, measurements, and calculations are necessary. A detailed description of the dosimetry and related methods and calculations is provided in Appendix A of the new ICNIRP Guidelines [1]. Very important is Specific Energy Absorption Rate (SAR, used below 6 GHz) or Absorbed Power Density (above 6 GHz). Measuring the SAR value is very complex because SAR depends on many factors, e.g., type of mobile phone, shape and size of the body or its part, dielectric properties of the tissue, position of the source, influence of surrounding objects, and frequency of electromagnetic waves [1,5,25].

In real conditions, there are several factors influencing the exposition to mobile phones, mainly:Time of active use of the phoneUsing of speaker mode or hands-free deviceDistance and visibility of the nearest mobile phone towerThe amount of mobile phone traffic in a certain place and timeThe model of mobile phone

### 2.3. Evaluated Parameters

The most important methods for evaluating the degree of damage to cells, chromosomes, and DNA are: the comet assay (measuring number of DNA strand breaks), gamma-H2AX detection, test of chromosomal aberration induction, micronucleus test, sister chromatid exchanges, examination of cell proliferation and cell cycle distribution, and detection of apoptosis [5,18,26]. Chromosomal and DNA damage should always be evaluated using more of the mentioned tests [5], and the results should be compared, combined, and interpreted carefully.

## 3. Main Technical Aspects of Laboratory Experiments

In laboratory experiments with live animals, many factors, such as stress, hormonal state, or seasonal effects, can play a role. In vitro studies examining the effects of radiofrequency radiation on cells appear easier to control, evaluate, and quantify. However, for the in vitro studies, it is also complicated to compare the results, especially when the experimental conditions are not carefully defined and described.

Experiments using different biological systems are difficult to compare because different cells and organisms may not respond to electromagnetic radiation in the same way.

Further, preparing and setting the technical part of the experiment (i.e., transmitter/transceiver), as well as evaluation and interpretation of the results, can be problematic, because the technologies and dosimetry may be outside the experience of the biomedical scientists [16].

Using a common mobile phone for the experiments implies problems with an accurate assessment of the dosimetry; thus, the experiments are neither well controlled nor repeatable (real mobile phone signal depends on many factors and is always unique for the given place and time). On the other hand, recent reviews by Panagopoulos [24,27] emphasize that the reaction to exposure is very different when comparing exposure to a uniform source (stable frequency and intensity) with exposure to a real mobile phone, with the highly and unpredictably variable signal. He states, based on the reviews of published peer-reviewed studies [24,27], even simulated mobile phone signals with regular pulsing will not have the same effect as a real, unpredictable mobile phone signal, i.e., that the variability makes the mobile phone signal more bioactive. In other words, the more variable the signal is, the more difficult it is for the organisms to adapt to it.

The results of the published research experiment reporting the health effects of radiofrequency radiation are often inconsistent and ambiguous in mutual comparison. It is often difficult to judge the differences in the studies, which wanted to repeat and test a previous study, and sometimes got discrepant results. However, what might appear inconsistent is indeed consistent with bimodal effects reported in hundreds of publications [8,16]. Generally, bimodal effects can be caused by the concentration of an agent, time of exposure, and many other parameters of the studied system. The results of the published research experiments show that the interaction of electromagnetic fields with DNA is very complex and depends on many factors, such as conditions of irradiation, cell type, and intensity and duration of the exposition [16].

### Development of Mobile Phone Technologies (Radiofrequency Sources)

The parameters of the telecommunication part of the experiments are very important and should be well controlled. Unfortunately, the radiofrequency radiation sources used in the studies are mostly either not well defined and controlled (a usual mobile phone), or do not simulate the unpredictable changes of the real mobile phone radiofrequency radiation (a vector signal generator). In recent years, new generations of mobile network technologies were introduced: 4G, using frequencies up to 8 GHz (mostly up to 2500 MHz [28]) and 5G [29], using frequencies below 6 GHz and up to 86 GHz (24.25–27.5 GHz for Europe [30,31]), i.e., new frequency bands and wider spectral bandwidth per frequency channel are assigned [32,33]. The future use of these higher frequencies and the need for a denser network of base stations is currently initiating debates about the influence of the 5G mobile infrastructure on organisms and human health. A new review by Karipidis et al. [17] found no confirmed evidence that low-level RF fields above 6 GHz are hazardous for human health. A review by Simkó and Mattsson [34] pointed out there were no consistent relationships between power density, exposure duration or frequency, and exposure effects in studies. Kostoff et al. [35] emphasize 5G technology can have effects other than only surface effects (on skin or eyes). It is obvious the results of older experiments, i.e., mobile phones/networks, do not apply to the new technologies due to new frequency bands, the number of antennas, modulation techniques, access methods, scheduling, etc., collectively called a radio access network.

## 4. Discussion

Despite the number of published works about genotoxicity and carcinogenicity of radiofrequency radiation, it cannot be definitely concluded if and to what extent this radiation is harmful (under normal circumstances and observing the safety limits) [1,36,37]. Harmful effects cannot yet be excluded, especially after the long exposure (many years) to low doses, which are typical for today’s population. Moreover, considering the quick development of mobile technologies, new experiments simulating and testing the effects of these new technologies are necessary. This implies that the effects of radiofrequency radiation will still be studied in the future, using both wider and better-defined in vivo studies, as well as laboratory experiments using ever more sensitive modern methods. In the large volume of published data, there is a key to finding the conditions that initiate DNA changes and to select suitable scientific methods for future studies [16].

The Organization for Economic Co-operation and Development (OECD) issued guidelines for the testing of genetic toxicity [38]. For basic principles of selecting and treating laboratory animals in in vivo experiments, the OECD guidelines for conduction of toxicity and carcinogenicity studies [39] and the National toxicology program specifications [40] provide detailed instructions. These include recommendations regarding the number of animals in study groups, details on housing and diet, and continuous (e.g., weight) and final (e.g., histopathology) evaluation. A “checklist” for a good quality study and publication is summarized in [20].

It is very important to clearly set, follow, note, and describe the conditions of the experiments. A methodology can be based on subjective and objective evaluation methods known from telecommunication technology for Quality of Service/Quality of Experience (QoS/QoE) evaluation [41,42].

Based on the literature review, we recommend the following points to be considered during the implementation and evaluation of experiments, Table 1.

Moreover, we emphasize that the dosimetry should always be clearly set and observed in the laboratory tests. This includes using a radiofrequency radiation source where the frequency, time, and intensity of the radiation, including placement of the radiation source, can be precisely set and controlled. To comply with the above-mentioned criteria—precisely defined radiofrequency radiation on the one hand, and unpredictable changes of the signal on the other hand—we suggest using the Software Defined Radio (SDR) approach available for 2G, 3G [44,45], 4G, and 5G [46] instead of a signal vector generator or undefined cell phone. The SDR transceiver enables the generation of high-frequency multichannel/wideband power signals in repeatable scenarios (recorded signal or artificial signal), with respect to timing, modulation methods, waveform, transmission power, and its time changes, etc., of real mobile phone radio channel parameters and for different xG mobile generations, unlike a standard signal generator. In order to be able to repeat the experiments in a controlled manner, it is necessary to use a standardized antenna adapter for radiating high-frequency power, i.e., an antenna structure for near/far EM field. The SDR enables the creation of a new base transceiver station (2G), Node B (3G), eNode B (4G), or gNode B (5G), which directly communicates with mobile phones, i.e., it can generate a real unpredictable mobile phone signal (i.e., signal changing in an irregular, unpredictable way), as suggested by Panagopoulos [24]. In addition, the SDR also enables the creation of a mobile phone phantom, i.e., specific hardware solution, e.g., OsmocomBB [44], with a well-defined mobile phone phantom (antenna, controlled radiofrequency signal radiation setup, etc., in standardized measurement environment, i.e., measurement cell), which can facilitate the reproducibility of future experiments. All necessary parameters of individual mobile technologies, including time intervals, frequency, modulation, intensity, etc., can be precisely controlled. Using the SDR and standardized measurement cell could substantially help to avoid problems of the replication of studies. In addition, one of the advantages of this approach is the possibility of sharing a specific scenario between laboratories in order to ensure the repeatability of the radiation source.

## 5. Conclusions

The paper reviews laboratory methods studying the effects of radiofrequency radiation on organisms, focusing on technical aspects of the experiments. As can be concluded from many works, the effects of radiofrequency radiation can differ under different conditions and settings. Therefore, we emphasize that the methodology should be clearly and precisely set to ensure the results can be verified and reproduced. Very important is the choice of a suitable source of radiation, here we recommend the SDR, which can simulate a real mobile phone signal in a controlled and repeatable way. The influence of radiofrequency radiation is a multidisciplinary topic and includes many fields, such as medicine, biology, toxicology, physics, electrical and electronic engineering, telecommunication, and statistics. The future research requires close cooperation of scientists from all these fields. Following the given recommendations should increase the overall quality of the experiments and publications and give the possibility to compare, reproduce, and verify the results.

## Figures and Tables

**Table 1 ijerph-18-12642-t001:** Recommendation for conditions of the laboratory experiments.

Parameter	Details that Need to Be Set Observed, Indicated and Described
Working place	1. Sample settings (placement, conditions, etc.) 2. Device setup (i.e., placement of devices with respect to the tested samples (animals or cells), use electrically non-conductive holders and cages, distance definition) 3. Scheme/block diagram
Conditions and treatment	1. Overall conditions (e.g., housing, temperature, light) 2. Treatment (e.g., feeding) of the exposed cells or live animals [40])
Details of RF radiation	1. Mobile technology (3G, 4G, 5G, etc., 3GPP Release number) 2. Source of radiation (signal generator, SDR) 3. Details of signal (frequency, signal shape, intensity, modulation, antennas, reflection/absorption components) 4. Mode of mobile phone operation (avoid standby mode, i.e., calling mode, transmission (communication) mode) 5. Comparison with allowable limits (e.g., FCC [25] limits for Maximum Permissible Exposure (MPE), ICNIRP [1] basic restrictions/reference levels, limits for controlled/uncontrolled environment/exposure, reference level for incident power density 40 Wm^−2^ for 2–6 GHz, etc.)
Specific dosimetry and SAR (or absorbed power density) evaluation	1. SAR setting and evaluation (below 6 GHz), using values 0.08 and 0.4 (whole body average), or 2, 4, 10, and 20 (local) Wkg^−1^ (exposure scenario, frequency range) 2. Absorbed Power Density evaluation (above 6 GHz), using values 20, 100 Wm^−2^ [1] 3. Comparison with allowable limits (e.g., SAR 1.6 W/kg [25])
Time of exposure	1. Duration of in vitro experiments typically up to 24 h 2. Duration of in vivo studies on animals usually days to, ideally, years. Intermittent exposure can be used 3. Cumulative exposure consideration [43]
Type of samples	1. Type of cells or animals 2. Exact specification, source
Sample size, groups of samples	1. Number of samples (i.e., large enough to provide statistically significant results), taking into account the type of samples and evaluated endpoints 2. Comparison with an unexposed or sham-exposed group [21]
Specific parameters of genotoxicity and DNA damage evaluation	1. Using more genotoxic indices when the cell and DNA damage are evaluated 2. Detailed description of the evaluation methods 3. Description of the detection limit 4. All final analyses and evaluations should be performed blind

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
