# Peer review of "Methodology of Studying Effects of Mobile Phone Radiation on Organisms: Technical Aspects"

_ijerph, 2021, doi:10.3390/ijerph182312642_

Round 1

Reviewer 1 Report

The introduction to the paper starts with a list of Possible negative effects of RF and gives a ref to Kesari et al. However, Kesari is only about male reproduction and discuss m ostly thermal effect. I think a broader ref would be better, for instance SCENIHR 2015.

Some lines further down comes almost the same list again and with a ref to Kesari et al. Both are not needed and should be rewritten as only one list with better refs.

1.1.1 sets out to deal with Laboratory studies in vivo but we have to wait till 1.5 to bring up the studies from NTP and Ramazzini. I think it would be better to deal with the in vivo first before bringing in the in vitro part ( 1.1.2). 

Under the heading 1.2 Experimental settings I am missing a mentioning about how important the cell incubator environment is. See further for instance
Hansson Mild K, Wilén J, Mattsson MO, Simko M. Background ELF magnetic fields in incubators: A factor of importance in cell culture work. Cell Biology International, 2009. doi: 10.1016/j.ceelbi.2009.04.004
Hansson Mild K, Mattsson MO, Hardell L. Magnetic fields in incubators a risk factor in IVF/ICSI fertilization? Electromagnetic Biology and Medicine, vol 22, pp. 51-53, 2003.

I also would like to see a note about not using a mobile phone in stand-by mode or any other use of a single phone for instance placed under a rabbit cage!

The paper deals with a problem of exposure to mobile phone or base station radiation and many, many published studies are lacking to cover the aspects that the authors bring up here.

With my minor remarks then paper should be accepted for publication.

Author Response

1. The introduction to the paper starts with a list of Possible negative effects of RF and gives a ref to Kesari et al. However, Kesari is only about male reproduction and discuss mostly thermal effect. I think a broader ref would be better, for instance SCENIHR 2015.

  • Thank you for the comments. The parts “Introduction” and “1.1 Types of Studies of Biological Effects” are rewritten.
  • The SCENIHR 2015 reference was added.

2. Some lines further down comes almost the same list again and with a ref to Kesari et al. Both are not needed and should be rewritten as only one list with better refs.

  • The parts “Introduction” and “1.1 Types of Studies of Biological Effects” are rewritten and references are added.

3. 1.1.1 sets out to deal with Laboratory studies in vivo but we have to wait till 1.5 to bring up the studies from NTP and Ramazzini. I think it would be better to deal with the in vivo first before bringing in the in vitro part ( 1.1.2). 

  • The section 1.5 about “Recent large in vivo studies” was moved after the 1.1.1 section about “Laboratory studies in vivo“ to 1.1.2 as suggested.

4. Under the heading 1.2 Experimental settings I am missing a mentioning about how important the cell incubator environment is. See further for instance
Hansson Mild K, Wilén J, Mattsson MO, Simko M. Background ELF magnetic fields in incubators: A factor of importance in cell culture work. Cell Biology International, 2009. doi: 10.1016/j.ceelbi.2009.04.004
Hansson Mild K, Mattsson MO, Hardell L. Magnetic fields in incubators a risk factor in IVF/ICSI fertilization? Electromagnetic Biology and Medicine, vol 22, pp. 51-53, 2003.

  • Thank you for the comments. The note about importance of conditions in the cell culture incubator environment was added, especially the ELF magnetic fields. The reference “Background ELF magnetic fields in incubators: ...” was added.

5. I also would like to see a note about not using a mobile phone in stand-by mode or any other use of a single phone for instance placed under a rabbit cage!

  • Thank you for interesting and useful hint. The remarks are added to Table 1.

6. The paper deals with a problem of exposure to mobile phone or base station radiation and many, many published studies are lacking to cover the aspects that the authors bring up here.

  • Thank you for the comment. The type of this paper is a Commentary, not a review or systematic review. Nevertheless, we added more references.

7. With my minor remarks then paper should be accepted for publication.

  • Thank you.

Reviewer 2 Report

Please answer my comments and suggestions listed in the attached file.

Author Response

Thank you for your comments.

1) Page 4, line 139: Please clarify the meaning of the sentence “Genetic toxicology study, comet assay and micronucleus assay, was also performed.” More descriptive variant could be the following: “Genetic toxicology study with analysis of the comet assay and micronucleus assay was also performed.”

  • The sentence was changed as suggested.

2) Page 4, lines 154-155: It seems that calling “the absence of the unpredictable changes of the radiofrequency signal” as a “disadvantage” contradicts the strategy of controlled experiments, where the technical parameters of the signal should be predicted and controlled in the standardized measurement environment. Please explain why the predictability of the changes in the signal is a disadvantage).

  • The explanation by the review of Panagopoulos 2019 is stated and reference to the next section of the paper was added.

3) Page 4, lines 172-174: Please explain why the use of the mobile phones in the experiments caused problems with dosimetry, control, and repeatability? Before on page 4, line 155 you mentioned that “the use of signal generator” is not good, but mobile phones also caused problems. Discussion about the SAR of mobile phones should include the reference to the Federal Communications Commission: https://www.fcc.gov/general/specific-absorption-rate-sar-cellular-telephones. Because FCC has estimated the limits for safe exposure to radiofrequency energy, the signal emitted by the mobile phone is predictably safe. What exactly means the effect you have mentioned on page 4 line 178 as an effect from “real unpredictable mobile phone signal”?

  • Problems with usual mobile phone were further explained. The point is that the real phone signal changes irregularly and unpredictably, it is influenced by many factors, specific for the given place and time. Thus if we want to repeat an experiment we are not able to get the same signal/exposure with a usual mobile phone.
  • Reference to Federal Communications Commission was added.
  • The term „real unpredictable mobile phone signal“ was further explained
  • The main point is, as is suggested in the reviews by Panagopoulos, the organisms may be able to adapt to the stressor, in this case the mobile phone signal, if it is stable or only slightly or regularly variable. Based on his review of published literature, he states the variability of the signal can be one of the crucial parameters causing biological effects.

4) Page 5, lines 187-199: How to find the numbered references (14-19) in the list of named References at the end of the paper? Such numbered references are mentioned in this sentence part: “…(14)) and 5G (15), using frequencies below 6 GHz and up to 86 GHz (24.25-27.5 GHz for Europe (16), (17)), i.e., new frequency bands and wider spectral bandwidth per frequency channel are assigned (18), (19).” Should we count 14 references from the beginning of the list?

  • The correct references were added.

5) Concerning the recommendation for conditions of the laboratory experiments you compared in Table 1, it could be very useful if you mention some limiting values for the RF radiation. The limits could be taken from your reference (WHO 2014) or FCC mentioned in my comment #3.

  • Some limiting values for the RF radiation were added with the recommendation to compare evaluated values with the limits given by WHO/FCC. References to the guidelines were added.

6) From the whole manuscript it is not clear whether the Software Defined Radio (SDR) exists and is available for the specialists. If SDR exists, please provide a reference. If it does not exist, then the importance of this commentary article is much lower and will be consideredas unsupported recommendations. You have written in Section 4. Conclusions: “we recommend the SDR, which can simulate a real mobile phone signal in a controlled and repeatable way.” Does the SDR exist or not?

  • The SDR exists and it works. In the laboratory of our department, we have upgraded the HW/SW to 5G. It is available for the reader in references, which were not well formatted. Now, 2G and 3G can be found as “Open Source Mobile Communications, OpenBTS“ reference and 4G, 5G as „OpenAir Interface“ reference